# Molecular Dynamics Simulation of the Influence of RDX Internal Defects on Sensitivity

**Pengmin Yan [1], Xue Zhao [1,\*], Jiuhou Rui [1], Juan Zhao [2], Min Xu [2] and Lizhe Zhai [1]**

1   State Key Laboratory of Explosion Science and Technology, Beijing Institute of Technology,
    Beijing 100811, China; 3120180265@bit.edu.cn (P.Y.); ruijiuhou@bit.edu.cn (J.R.); 3220190166@bit.edu.cn (L.Z.)
2   Analysis and Testing Center, Xian Modern Chemistry Research Institute, Xi'an 710065, China;
    zcjsheep204@163.com (J.Z.); naturemin2001@163.com (M.X.)
\*   Correspondence: zhaoxue@bit.edu.cn

**Abstract:** The internal defect is an important factor that could influence the energy and safety properties of energetic materials. RDX samples of two qualities were characterized and simulated to reveal the influence of different defects on sensitivity. The internal defects were characterized with optical microscopy, Raman spectroscopy and microfocus X-ray computed tomography technology. The results show that high-density RDX has fewer defects and a more uniform distribution. Based on the characterization results, defect models with different defect rates and distribution were established. The simulation results show that the models with fewer internal defects lead to shorter $N-NO_2$ maximum bond lengths and greater cohesive energy density (CED). The maximum bond length and CED can be used as the criterion for the relative sensitivity of RDX, and therefore defect models doped with different solvents are established. The results show that the models doped with propylene carbonate and acetone lead to higher sensitivity. This may help to select the solvent to prepare low-sensitivity RDX. The results reported in this paper are aiming at the development of a more convenient and low-cost method for studying the influence of internal defects on the sensitivity of energetic materials.

**Keywords:** RDX; sensitivity; recrystallization; molecular dynamics





## 1. Introduction

Energetic materials play a significant role in both the military and in civilian life. However, the internal defects such as cracks, voids, and inclusions seriously increase the sensitivity of energetic materials, causing difficulties in subsequent use [1–4]. The defects in the crystal are different and complex. Therefore, scientifically studying the influence of internal defects on sensitivities of energetic materials is still a challenge.

Knowing the information of internal defects will help researchers to further learn about the specific effects of defects on sensitivity and improve the safety performance of energetic materials. People have carried out lots of characterization and sensitivity experiments on energetic materials with internal defects [5–12]. These researchers revealed the connection between the characterization results and internal defects, establishing the foundation for further research. Due to the inability to establish the corresponding molecular dynamics model for the column defects caused by charge, the influence of crystal defects and charge defects on sensitivity cannot be quantitatively characterized. Therefore, the crystal internal defects are simulated separately in order to understand their influence on sensitivity.

In this paper, using RDX as a case, molecular dynamics simulation is proposed to study the influence of the sensitivity of internal defects in RDX crystals combined with internal defects characterization. Three non-destructive characterization methods, optical refraction matching, Raman spectroscopy, and micro focus X-ray computed tomography technology (μ-CT) were performed on RDX samples of two qualities. Based on the results of the characterization, the defect rates, types, and distribution were obtained, and the

corresponding defect models were established. The simulation results were analyzed, and two parameters were taken as a criterion of relative sensitivity of different defect models. According to the simulation results, the defect model which has a significant impact on sensitivity will be found. Then, the RDX crystal can be optimized in a targeted manner. The molecular dynamics simulation was used to simulate many properties before, such as mechanical properties, prediction of crystal morphology, and so on, and the results are all in good agreement with experiments [13–18]. The simulation method provides a safe, convenient, fast, and low-cost way to study the influence of internal defects on the sensitivity of RDX crystals. The method could also be used to study the influence of internal defects on the sensitivity of other energetic materials and crystals and provide a reference for subsequent optimization.

## 2. Materials and Methods

The materials used in this study were raw RDX (purchased from Gansu Yinguang Chemical Co., Ltd., China) and high-density RDX, which was recrystallized in our laboratory from raw RDX. Raw RDX was dissolved in cyclohexanone at 120 centigrade, then cooled naturally to room temperature with stirring speed of 100 r/min.

The X-ray diffraction was taken by an X-ray single crystal diffraction (SMART APEX, AXS Brook, Germany).

The internal defects were observed by optical spectroscopy (DM4M, Leica Microsystems, Germany).

The defect distribution was characterized by Raman spectroscopy (inVia Reflex, Renishaw, UK).

The defect rate was characterized by a high-resolution computed tomography system (YXLON, Germany).

The RDX crystal structure was taken directly from the experimental data [19], which belongs to an orthorhombic crystal system with Pbca symmetry and Z = 8. The lattice parameters of RDX are a = 13.182 Å, b = 11.574 Å, c = 10.709 Å, $\alpha = \beta = \gamma = 90°$.

All the simulations of the defect models were performed with the Forcite module of the Materials Studio 17.0 software (Accelrys, San Diego, CA, USA). The accuracy of the COMPASS (condensed-phase optimized molecular potential for atomistic simulation study) force field in nitroamine explosives was validated. In the COMPASS force field, the molecular parameters of nitroamine explosives are in good agreement with the experimental results [20]. The simulation box was the RDX primitive cell expanded into a $3 \times 3 \times 3$ supercell, with a total of 4536 atoms and 216 RDX molecules. After geometry optimization, the equilibrium simulation was conducted for 1ns in the NPT ensemble. The equation of motion was integrated with a time step of 1 fs. The trajectory data were collected every 10 time steps. The temperature was maintained at 298 K with the aid of an Andersen thermostat [21]. The pressure was maintained with the aid of a Parrinello barostat [22]. The electrostatic forces were evaluated by the Ewald summation approach and the van der Waals interactions were truncated with a smoothed spherical cut-off of 12.5 Å.

## 3. Results

### 3.1. Optical Refraction Matching

Two samples were placed in refractive index matching liquid and observed by the optical microscope. Figure 1 shows the optical micrographs of two different quality RDX. (a) is high-density RDX, (b) is raw RDX. The images have consistent magnification. Because the refractive index does not match between internal defects and the sample, these internal defects will appear as black spots in the optical micrographs. It can be seen that the light transmittance of the high-density RDX is significantly higher than that of the raw RDX. Obviously, the high-density RDX has fewer defects.

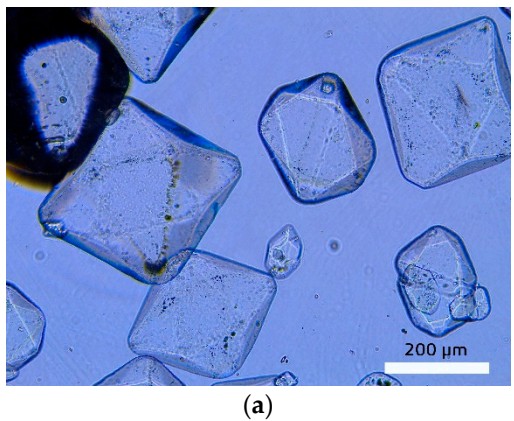
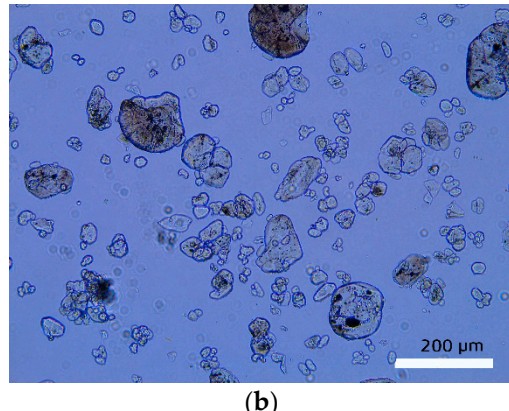

(**a**)  (**b**)

**Figure 1.** Optical micrographs of (**a**) high-density RDX and (**b**) raw RDX.

### 3.2. Raman

The Raman spectra of high-density RDX and raw RDX are illustrated in Figure 2. It can be seen that two Raman spectra are consistent. The high-density RDX did not undergo a phase transition. The sharpest peaks all appear at 883 cm$^{-1}$. Choosing 883 cm$^{-1}$ as the characteristic peak, FWHM of the characteristic peak and the relative standard deviations (RSDs) of FWHM are shown in Table 1.

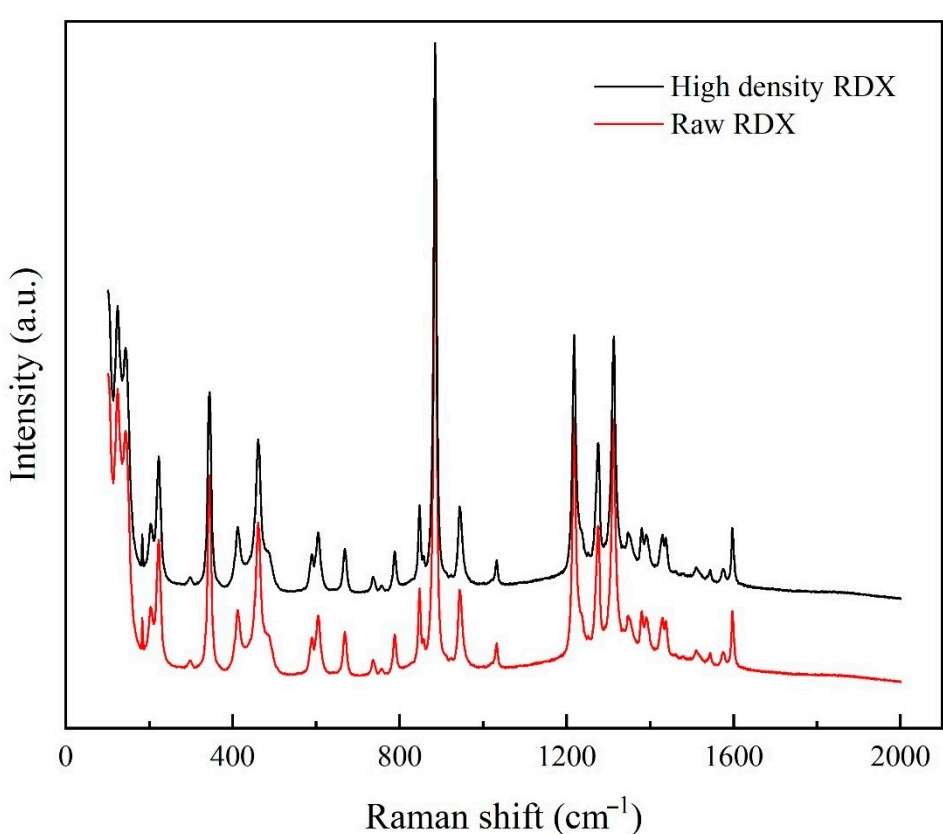

**Figure 2.** Raman spectra of high-density RDX and raw RDX.

**Table 1.** FWHM and its relative standard deviation (RSD) of high-density RDX and raw RDX.

| Sample | FWHM/cm$^{-1}$ | RSD/cm$^{-1}$ |
|---|---|---|
| High-density RDX | 8.26 | 0.010 |
| Raw RDX | 8.31 | 0.016 |

When the atoms are arranged completely according to the theoretical structure, the phonon coherence length is the longest, and the phonon wave propagation in the crystal is not affected. There is no energy loss. The central vibration of the Brillouin zone is the Raman spectral vibration. The line width of Raman characteristic peaks is 0. When there are defects inside the crystal, the phonon wave propagation in the crystal is attenuated, the phonon energy will lose, and vibrations not in the Brillouin zone will appear, which will broaden the line width. That is to say, as the defects in the crystal increase, the FWHM of the Raman characteristic peak will broaden. The relative standard deviation (RSD) is the ratio of the standard deviation to the arithmetic mean value, which can characterize the dispersion of the data based on the arithmetic mean value. Therefore, the RSD of FWHM can characterize the distribution of internal defects and the smaller the RSD, the more uniform the defect distribution. It can be seen from Table 1 that the FWHM and RSD of high-density RDX are smaller than raw RDX, indicating that high-density RDX crystal has fewer internal defects and more uniform distribution than that of raw RDX. This is consistent with the results of the optical refraction matching experiment.

*3.3. Micro-CT*

μ-CT is a non-invasive and non-destructive imaging technique that can obtain images without destroying the sample. Figure 3 is a μ-CT scan cross-section of high-density RDX. The detail recognition ability is less than 3 μm (refractor)/0.5 μm (transmission). The dimensional accuracy is less than 5 μm. Due to the small particle size of raw RDX, a clear μ-CT scan figure cannot be obtained.

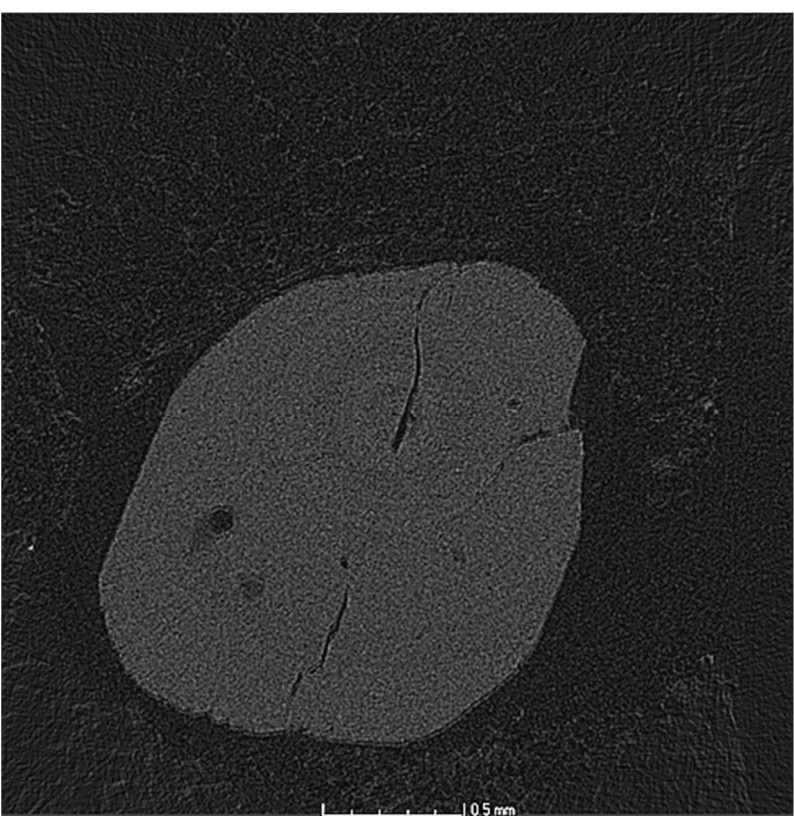

**Figure 3.** μ-CT scan cross-section of high-density RDX.

As is illustrated in Figure 3, there are holes and cracks in the crystal. The diameter of the crystal is about 1200 μm, and it is not a single crystal. The diameter of the holes is from 10 to 100 μm, and the length of the crack is from 10 to 400 μm. The holes may be voids or inclusions. The cracks may be caused by internal stress which is generated by internal defects.

The gray value of the μ-CT image can indirectly reflect the density distribution of the sample. The dark shadow area with lower gray value is low-density area, and the white shadow area is high-density area. If the gray value was distributed concentratedly, the density distribution of the crystal would be more uniform, and the internal defects of crystal would be fewer. The gray value of the effective region in Figure 3 was calculated and 144,000 pixels were obtained in total. As the background grayscale, pixels with gray values less than 10 were deducted and 140,810 pixels were left after deducting. Taking the interval as 10, the gray value distribution of high-density RDX is shown in Table 2.

**Table 2.** The number and proportion of the gray value in μ-CT scan cross-section.

| Grey Value | Pixel Number | Proportion |
|---|---|---|
| (10,20] | 9352 | 6.64% |
| (20,30] | 15,163 | 10.77% |
| (30,40] | 20,737 | 14.73% |
| (40,50] | 35,719 | 25.37% |
| (50,60] | 37,837 | 26.87% |
| (60,70] | 17,953 | 12.75% |
| (70,80] | 3693 | 2.62% |
| (90,100] | 337 | 0.23% |
| (100,110] | 17 | 0.01% |
| (110,120] | 2 | 0.00% |

The gray value is distributed between 10 and 120. The number of pixels with gray values above 100 can almost be ignored. The proportion of pixels with gray values below 70 is as high as 97.13%, and those between 40 and 60 account for more than 50%. The gray value distribution is highly concentrated, indicating that the density distribution of high-density RDX crystals is relatively uniform, and the submicron defects are few.

The external energy absorbed by the explosive is transformed into heat energy, which is concentrated in the local area or in some points. The concentrated energy can make the temperature rise rapidly, forming the so-called "hot spot". Since the minimum size of the hot spot is 3–5 μm [23], which is consistent with the resolution of the μ-CT image, the crystal defects that cannot be resolved in the image cannot form the hot spots. Calculating with the grid method, the area of the holes and cracks in the figure accounted for about 1.5–2% of the entire crystal area. Due to it being a cross-section of the crystal, it can be considered that the true defect rate of defects, which can affect the RDX sensitivity, is less than 1.5%.

### 3.4. Molecular Dynamics Simulation

As can be seen from the above results, the internal defects in high-density RDX are less than raw RDX and is distributed more evenly. Four different models were established according to the characterization results. In addition to the defect-free perfect model there was (a) the vacancy model, (b) with six RDX molecules evenly deleted and the vacancy model and (c) with four RDX molecules concentratedly deleted, which were established by being based on raw RDX. Then based on high-density RDX, vacancy model (d) with four RDX molecules evenly deleted. The defect models are shown in Figure 4, and the specific information of these models is shown in Table 3.

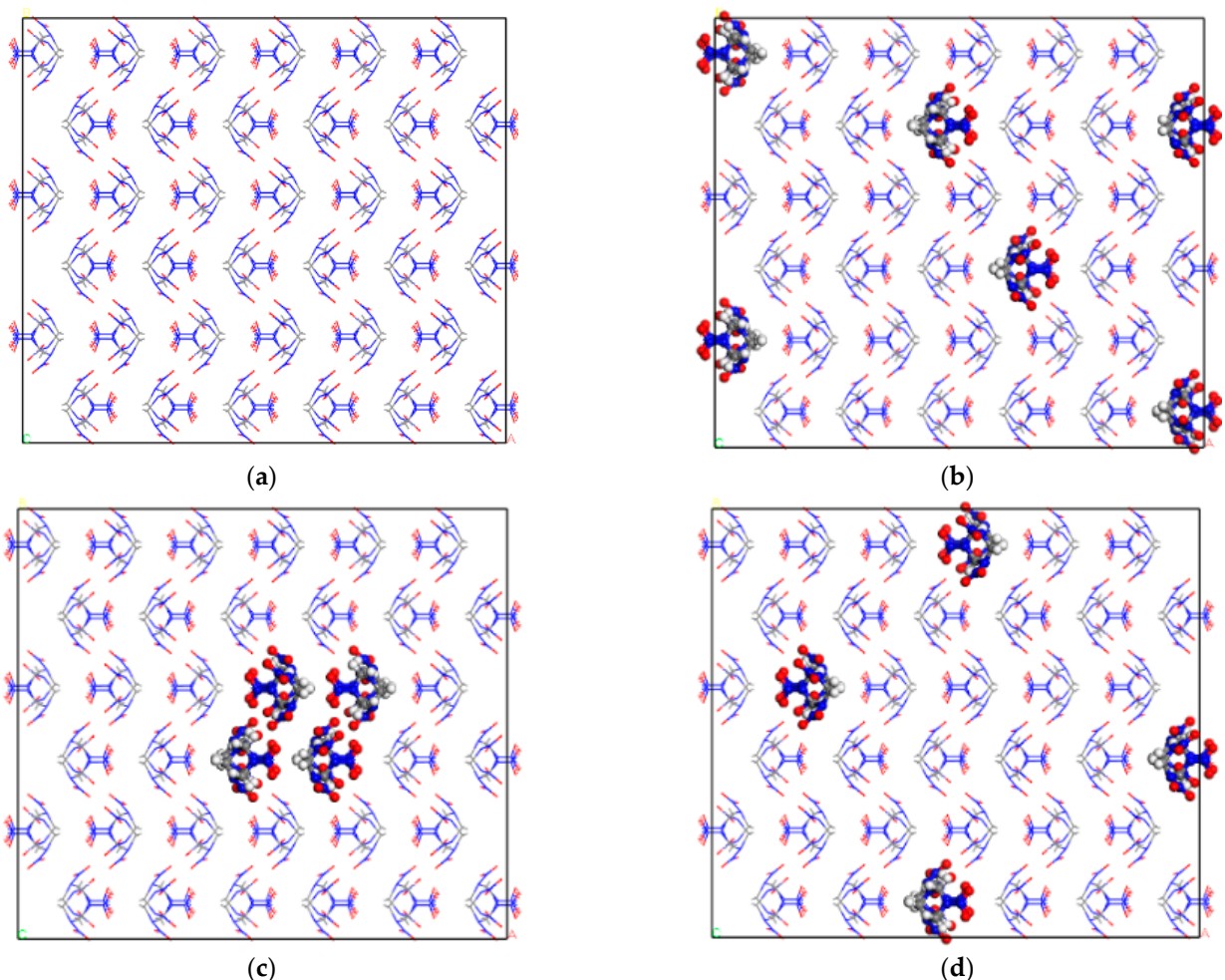

**Figure 4.** Different defect models: (**a**) defect-free model, (**b**) 6 molecules vacancy model, (**c**) 4 molecules vacancy model, and (**d**) 4 molecules vacancy model.

**Table 3.** Defect types, defect rates and corresponding crystals of each model.

| Model | Defect Type | Defect Rates | Crystal |
|---|---|---|---|
| (a) | None | 0 | - |
| (b) | Vacancy | 2.78% | Raw RDX |
| (c) | Vacancy | 1.85% | Raw RDX |
| (d) | Vacancy | 1.85% | High density RDX |

Table 4 shows the cohesive energy density (CED), N–NO$_2$ maximum bond length (L$_{max}$), and the activation bonds ratio of each model. CED is the energy required for 1 mol condensate per unit volume to overcome the intermolecular force and become gaseous. In other words, the smaller the CED, the smaller the energy required to change from crystalline to gaseous state. So, to some certain extent, CED can be considered as a theoretical criterion for relative thermal sensitivity. It is pointed out in the literature that the N–NO$_2$ bond is the pyrolysis or initiation bond of nitroamine explosives [24], and the molecules with the largest bond length are the activated molecules, which are most likely to cause decomposition and initiation. The activation bonds ratio is defined as the ratio of the activated bond lengths to the total bond length distribution. The activated bonds are compared to the defect-free model (a), the activated bond is defined as a bond longer than 1.563 Å. Therefore, the maximum bond length of the N–NO$_2$ bond and activation bonds ratio can be one of the theoretical criteria for the relative sensitivity of nitroamine explosives.

**Table 4.** Cohesive energy density (CED), maximum N–NO$_2$ bond length and activation bond ratio of each model.

| Model | CED/J·cm$^{-3}$ | Lmax/Å | Activation Bonds Ratio |
|---|---|---|---|
| (a) | 754.5 | 1.563 | 0 |
| (b) | 676.8 | 1.579 | 0.0018‰ |
| (c) | 687.2 | 1.575 | 0.0014‰ |
| (d) | 688.6 | 1.572 | 0.0010‰ |

It can be seen from Table 4 that the CED of the defect-free model is the largest and L$_{max}$ is the shortest. The CED of the other models are smaller and the L$_{max}$ are longer than that of the defect-free model. That is, the presence of defects will increase the crystal sensitivity, which is consistent with the actual situation. Model (b) has the highest defect rate, correspondingly CED is the smallest, L$_{max}$ is the longest, (c) has lower CED and longer L$_{max}$ than (d). The activation ratio is corresponding with the CED and L$_{max}$, model, and (b) has the most activated bonds, followed by (c) and (d). These data indicate that the higher the defect rate and the more uneven the defect distribution, the higher the sensitivity, which is also consistent with the actual situation. Though point defects cannot form a hot spot, the existence of point defects will affect the corresponding parameters, and the change of these parameters will lead to the initiation bonds becoming easier to decompose, and the energy accumulates faster at hot spots, increasing the sensitivity.

The above results show that the sensitivity of the fewer defective models is lower than others. The characterization experiment results show that high-density RDX has fewer internal defects. The research group also did an experiment, in which the results found that the sensitivity of high-density RDX crystal was lower than that of raw RDX [25], proving that the result of simulation calculation could be used as a criterion of relative sensitivity, to a certain extent.

Besides the vacancy, inclusion is another important cause of hot spots. Therefore, it is necessary to discuss the influence of different solvents doping models on crystal sensitivity. The models doped with four H$_2$O, cyclohexanone, acetone, NMP, butyrolactone, propylene carbonate, DMSO, and DMF molecules were established respectively. The H$_2$O doping model is shown in Figure 5 and the other models are all the same as it. The simulation results are shown in Table 5.

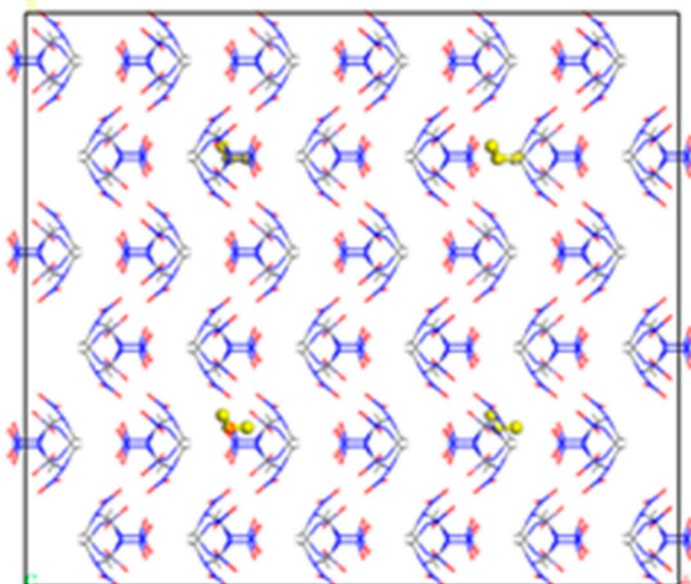

**Figure 5.** The H$_2$O doping defect model.

**Table 5.** CED, maximum N–NO$_2$ bond length, and activation bonds ratio of solvent doping models.

| Doped Molecular | CED/J·cm$^{-3}$ | Lmax/Å | Activation Bonds Ratio |
|---|---|---|---|
| H$_2$O | 703.3 | 1.600 | 0.0018‰ |
| cyclohexanone | 692.1 | 1.589 | 0.0015‰ |
| acetone | 564.4 | 1.646 | 0.1191‰ |
| NMP | 695.1 | 1.583 | 0.0012‰ |
| butyrolactone | 693.9 | 1.591 | 0.0014‰ |
| propylene carbonate | 608.5 | 2.162 | 1.5440‰ |
| DMSO | 689.2 | 1.593 | 0.0011‰ |
| DMF | 695.1 | 1.589 | 0.0009‰ |

Obviously, the results of models doped with acetone and propylene carbonate are quite different from the other models. The activation bonds ratios of acetone and propylene carbonate are a hundred and a thousand times more than the other models, indicating that the percentage of bonds in these two models that are activated is very high. Especially the N–NO$_2$ bond length distribution of the model doped with propylene carbonate, which is shown in Figure 6. There is no N–NO$_2$ bond distribution between 1.570 Å and 2.000 Å, and while the bond length distribution appears between 2.000 Å and 2.160 Å again, this phenomenon does not appear in the other models. The bond length distribution is not continuous. This shows that when the propylene carbonate doping activates RDX molecules, the activation of molecules is not continuous. The activated molecules are excited to a high-energy state, while other molecules are still in a steady state. There appears to be an energy barrier in RDX molecules.

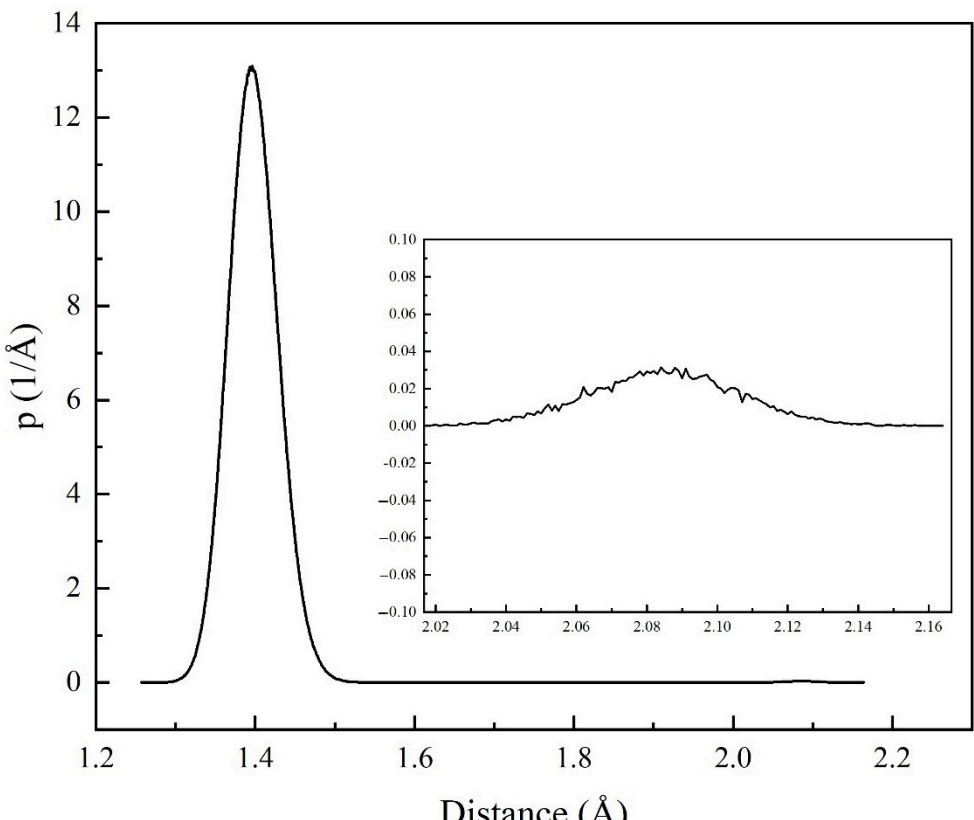

**Figure 6.** N–NO$_2$ bond length distribution of propylene carbonate molecular doping model.

Except for acetone and propylene carbonate, the other doping models have similar CED, L$_{max}$ and activation bonds ratio, indicating that there is no significant difference in the sensitivity of recrystallization with these solvents when other factors are not considered.

The influence of different solvent doping on the CED, $L_{max}$, and activation ratio is still unclear, which requires further theoretical analysis and simulation research.

## 4. Discussion

Limited by computer power and crystal detection technology, these defect models are very crude. Defect types are not complete, and the model types are too singular, which cannot reflect true internal defects. However, it is of great importance to establish the quantitative defect model based on the crystal internal defect characterization results. Due to the danger of energetic materials, computer simulations are safer and more convenient. Besides, the sensitivity experiments require the preparation of energetic materials into charge, during which the charging process and additives also have affection. It is difficult to know specifically the effect of crystal defects on sensitivity. By computer simulation, the influence of different defects and their combination on sensitivity can be analyzed concretely without the interference of external factors.

In terms of the criterion of sensitivity, a longer $N-NO_2$ bond length means less activation energy required for bond cracking, so the maximum bond length is a reasonable criterion of sensitivity. The number of activated bonds also indicates whether the model is more prone to initiation decomposition. From the definition of CED, the thermal sensitivity of energetic materials is related to it. However, just as the simulation results show above, different solvent doping has differing influences on CED, and the rule is not clear. The relation between CED and thermal sensitivity needs further experiments for verification.

In summary, molecular dynamics simulation has the advantages of safety, convenience, high efficiency, and low cost in studying the effects of internal defects on crystal properties. Combining the characterization results with modeling, the properties of target crystal can be simulated. This is of guiding significance to the optimization of energetic material crystals.

## 5. Conclusions

In order to study the influence of internal defects on the crystal sensitivity of energetic materials, the molecular dynamics simulation method combined with the characterization experiment was proposed to compare the relative sensitivity of the crystal. According to the characterization results of different quality RDX, the defect models used in the simulation were established, and the reliability of the simulation results was verified by combining the previous experimental results of the research group. The method of molecular dynamics simulation can establish defect models qualitatively and quantitatively based on the characterization results of internal defects and can compare the influence of different defects on the crystal sensitivity of energetic materials. This method can safely and efficiently discover the defects that have a greater impact on sensitivity, providing a direction for subsequent targeted optimization of energetic materials. This method could not only be used to study the internal defects of RDX but also of other energetic materials. Even so, the factors that affect the crystal sensitivity of energetic materials in preparation are very complicated. The results of molecular dynamics simulation still need experiments to optimize and verify.

**Author Contributions:** Conceptualization, P.Y. and X.Z.; formal analysis, P.Y.; investigation, P.Y. and L.Z.; visualization, P.Y. and L.Z.; writing—original draft, P.Y.; project administration, J.R., X.Z.; supervision, X.Z.; writing—review & editing, X.Z.; funding acquisition, X.Z.; data curation, J.Z. and M.X.; resources, J.Z. and M.X. All authors have read and agreed to the published version of the manuscript.

**Funding:** This research received no external funding.

**Data Availability Statement:** Data available on request due to restrictions e.g., privacy or ethical.

**Acknowledgments:** Thanks to all members of the research group for their support.

**Conflicts of Interest:** The authors declare no conflict of interest.

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
