# Peer review of "Molecular Dynamics Simulation of the Influence of RDX Internal Defects on Sensitivity"

_crystals, doi:10.3390/cryst11040329_

Round 1
Reviewer 1 Report
The title of your paper is misleading: 1) it is worth to precise that it is a molecular dynamics simulation, 2) please define explicitly that you mean “shock” sensitivity, and 3) may be it is worth to omit “Based on Characterization » else describe all your techniques used to characterize the defect structure.
In the text, you say nothing about the way you impact your RDX specimen and it is not clear what kind of shock sensitivity parameter you simulate.
Variation of Lmax in Table 4 seems to be too small to affect the sensitivity. No use to discuss the effect of this parameter.
It is known [1] that the critical detonation diameter and a few other shock sensitivity parameters correlate well with the specific surface area of grains or pores, which I suppose is much higher than that of internal defects you consider (I would agree with you if you mean cast explosives, but apparently it is not the case). Please try to justify the importance of the inner crystal defects you study. Consequently, it seems worthy to modify your statement “The internal defect is a crucial factor which could significantly influence the energy and safety properties of energetic materials”.
Finally, there are a few misprints.
[1] - B.A. Khasainov, B.S. Ermolaev, H.-N. Presles, P. Vidal. On the effect of grain size on shock sensitivity of heterogeneous high explosives. Shock Waves (1997) 7:89–105
Reviewer 2 Report
A very valuable addition to the literature on the sensitivity of RDX.
Perhaps the authors could include a reference on the most recent and comprehensive report on explosives sensitivities (including RDX):
Energetic Materials Encyclopedia, Volumes 1 – 3, 2nd edn., T. M. Klapötke, De Gruyter, Berlin / Boston, 2021.
Author Response
Point 1:Perhaps the authors could include a reference on the most recent and comprehensive report on explosives sensitivities (including RDX):
Energetic Materials Encyclopedia, Volumes 1 – 3, 2nd edn., T. M. Klapötke, De Gruyter, Berlin / Boston, 2021.
Response 1: It will be added. Thank you very much for your recommendation.
Reviewer 3 Report
The paper attempt is to connect the presence of defects, such as voids, cracks, vacancies or inclusions in crystal of of a common military explosive, or energetic material, such RDX with the sensitivity of the explosive, that is the amount of energy required to initiate the explosion. This is an important property of the explosive. A safer explosive is less sensitive and will not explode if accidentally dropped or mishandled. A large research effort is devoted to correlate the defect content with sensitivity (see for instance the reported publications ).
In the paper raw or recrystallised RDX crystals are characterised by optical microscopy, by computed x-ray tomography, by Raman scattering. The authors also simulate by molecular dynamic RDX crystals with different types of defects, such as vacancies, or inclusions of some molecules of solvent in to the RDX lattices. In this way they calculate cohesive energy density (CED) and N-NO2 maximum bond length, and declare that there is a good correlation between lower CED and larger bond length with explosive sensitivity.
This is what I tried to understand from the difficult reading of the manuscript.
The paper reports interesting results but, mainly due to the poor quality of the English language, it is not easy to understand the paper content. Moreover some assumptions appear not justified.
For instance: by Raman spectra of row and recrystallised RDX seem very similar. The author note that the relative standard deviation (RSD) of Raman peak at 883 cm-1 of raw and recrystalised RDX assume quite different values. Is due to crystal disorder? How this can be correlated to the RDX sensitivity
By micro CT the author show cracks that are probably at the origin of a higher sensitivity but later they consider that “the crystal defects that cannot be resolved in the mCT image can be considered to have no effect on the crystal sensitivity”. But in a successive paragraph they simulate the crystal lattice and conclude that point defects enhance the RDX sensitivity.
Further they correlate RDX sensitivity to N-NO2 bond “It is pointed out in the literature that the N-NO2 bond is the pyrolysis or initiation bond of nitroamine explosives[22]” but reference 22 is not a publication, nor is available.
For these reason I recommend a careful rewriting of the paper and I cannot recommend the publication of the paper in the present form
Desensitization of high explosives by encapsulation in metal-organic frameworks By: Kim, Eun-Young; Hong, Do-Young; Han, Mingu; et al.CHEMICAL ENGINEERING JOURNAL Volume: 407 Article Number: 127882 Published: MAR 1 2021
DISLOCATION SLIP SYSTEMS IN PENTAERYTHRITOL TETRANITRATE (PETN) AND CYCLOTRIMETHYLENE TRINITRAMINE (RDX) By: GALLAGHER, HG; HALFPENNY, PJ; MILLER, JC; et al.
PHILOSOPHICAL TRANSACTIONS OF THE ROYAL SOCIETY OF LONDON SERIES A-MATHEMATICAL PHYSICAL AND ENGINEERING SCIENCES Volume: 339 Issue: 1654 Pages: 293-303 Published: MAY 15 1992
Round 2
Reviewer 1 Report
In reference [1] of my former comment, it is shown that sensitivity is governed by a specific surface area of all kinds of defects. Therefore, it is difficult to agree with your following statement in section 1:
“However, due to the complex causes of internal defects, and sensitivity experiment needs to prepare energetic materials into grains, introducing other influencing factors, it is difficult to know which kind of defects has a greater impact on the sensitivity only with experiments, and to reduce the effect of defects on sensitivity according to the information of internal defects.”
Once more, it is not clear what is the kind of impact on your explosive is simulate. At least try to explain briefly, what COMPASS serves for.
Reviewer 3 Report
This second version of the paper is considerably improved and I consider the publication acceptable after some minor changes.
I suggest to include in the paper the answers of the authors to my previous comments, or at least part of them.
Specify what is a “hot spot” of the explosive, say that defects of smaller dimension, not visible by microCT, cannot be considered as hot spots.
Still some ambiguity remains: if the reason of the increased RDX sensitivity is the presence of hot spots, that is defects larger than a few microns, why do you correlate the RDX sensitivity to point defects, as in the molecular dynamic simulation?
I received publication [22] form the authors, but it is not publicly available and is written in Chinese.
Fig3 does not show visible contrast changes as stated in table 2. Probably the contrast in the image can be enhanced.
